# Structural and Functional Aspects of the Spleen in Molly Fish *Poecilia sphenops* (Valenciennes, 1846): Synergistic Interactions of Stem Cells, Neurons, and Immune Cells

**DOI:** 10.3390/biology11050779

**Published:** 2022-05-20

**Authors:** Ramy K. A. Sayed, Giacomo Zaccone, Gioele Capillo, Marco Albano, Doaa M. Mokhtar

**Affiliations:** 1Department of Anatomy and Embryology, Faculty of Veterinary Medicine, Sohag University, Sohag 82524, Egypt; ramy.kamal@vet.sohag.edu.eg; 2Department of Veterinary Sciences, University of Messina, 98168 Messina, Italy; zacconegiacomo@gmail.com (G.Z.); gioele.capillo@unime.it (G.C.); 3Institute for Marine Biological Resources and Biotechnology (IRBIM), National Research Council (CNR), Section of Messina, 98100 Messina, Italy; 4Department of Chemical, Biological, Pharmaceutical and Environmental Sciences, University of Messina, 98166 Messina, Italy; 5Department of Cell and Tissues, Faculty of Veterinary Medicine, Assuit University, Assiut 71526, Egypt; doaa@aun.edu.eg

**Keywords:** melanomacrophages, IL-1β, lymphocytes, leukocytes, SOX9, stem cells, immune response, defense mechanisms, Poeciliidae, bony fishes

## Abstract

**Simple Summary:**

Fish are free-living organisms who are dependent on their innate immune system for survival. This system is fundamental for the defense mechanisms against invading pathogens. The head kidney, spleen, and thymus are the main immune organs in fish, albeit with slightly different roles between species. This study aims to highlight the cellular components of the spleen in the molly fish (*Poecilia sphenops*) by providing some background on their potential role in the immune system of fish. The splenic parenchyma of molly fish is comprised mainly of red pulp intermingled with clusters of white pulp cells. The main cells identified in this study included: erythrocytes, neurons, dendritic cells, stem cells, epithelial reticular cells, lymphocytes, ellipsoids, and granular leukocytes (eosinophils, basophils, neutrophils), and macrophages. The current study demonstrated a variety of immune cell types with specific functions in the spleen of molly fish. The strong positive immunoreactivity of the cellular constituents of the spleen for APG-5, IL-1β, NF-κB, and TGF-β suggests its critical role in immunity.

**Abstract:**

In fish, the spleen is the prime secondary lymphoid organ. It has a role in the induction of adaptive immune responses, in addition to its significance in the elimination of immune complexes. This study was conducted on 18 randomly obtained adult molly fish (*Poecilia sphenops*) of both sexes using histological, immunohistochemical, and ultrastructural studies to highlight the cellular components of the spleen and their potential role in the immune system. The spleen of molly fish was characterized by the presence of well-distinct melanomacrophage centers, and other basic structures present in higher vertebrates including red and white pulps, blood vessels, and ellipsoids. Some mitotic cells could also be identified in the red pulp. Mast cells with characteristic metachromatic granules could be seen among the splenic cells. Rodlet cells were randomly distributed in the spleen and were also observed around the ellipsoids. The white pulp of the spleen expressed APG5. The expressions were well distinct in the melanomacrophages, leukocytes, and macrophages. Myostatin was expressed in leukocytes and epithelial reticular cells. IL-1β showed immunoreactivity in monocytes and macrophages around the ellipsoids. NF-κB and TGF-β were expressed in macrophages and epithelial reticular cells. Nrf2 expression was detected in stem cells and rodlet cells. Sox-9 had a higher expression in epithelial reticular cells and stem cells. The high frequency of immune cells in the spleen confirmed its role in the regulation of both innate and adaptive immunity, cell proliferation, and apoptosis.

## 1. Introduction

The immune systems of fish and higher vertebrates are physiologically similar, except certain differences. Fish are free-living organisms and they rely on their innate immune system for survival, in contrast to higher vertebrates [1,2]. Nonspecific immunity is a basic defense mechanism in fish, and it is very important for the acquired immune response and homeostasis through the receptor proteins system [3,4].

The fish immune system is constituted of immunological components that are diverse and interdependent. It is essential for the defense mechanisms against invading pathogens [5,6]. One of the characteristic features of the immune system is that every component has its inherent protective value and is related to a satisfactory immune response [7,8,9].

The spleen, head kidney, and thymus are regarded as the main immune organs in fish, albeit with slightly different roles among species. The spleen is the predominant erythropoietic tissue in many fish species including elasmobranchs (sharks, rays), holocephalans (rabbitfish: Chimaera), and a few teleosts (Perca, Scorpaena) [10,11,12]. An extensive network of cellular elements exists mainly in the spleen and kidney for trapping blood-borne substances; however, other tissues such as the liver and heart are also engaged in some species. In the spleen and kidney, aggregations of lymphocytes and macrophages, with their capability of mounting an immunological response, are located close to antigen trapping sites and are often associated with melanomacrophage centers (MMCs) [13].

*Poecilia sphenops* (Valenciennes, 1846) is a freshwater fish species commonly named molly fish. They are natural inhabitants of freshwater streams and the coastal brackish water of Mexico and Colombia and mostly occur in swarms below the floating vegetation as they feed principally on algae and other herbal resources, and have been marked in many countries around the world through the aquarium fish trade [14]. Because of their higher birth size, growth rate, reproduction, and brood number, mollies are categorized as one of the most widely consumed fish. Moreover, this fish is a viviparous species [15,16]. The morphology of the immune system shows distinct variations among fish species. Little data are available regarding the general structure of their spleen; therefore, this study aimed to highlight the cellular components of the spleen in molly fish (*Poecilia sphenops*) by providing some background on their potential role in the immune system of fish.

## 2. Materials and Methods

### 2.1. Sample Collection

This study was conducted on 18 randomly obtained adult molly fish (*Poecilia sphenops*) of both sexes. The fish were purchased from an ornamental fish shop and had a standard length of 4.20 ± 4.0 cm, and a body weight of 10.60 ± 1.70 g. Healthy fish were acclimated in the laboratory for two weeks in aerated water tanks and a natural light/dark cycle. Dissolved oxygen and temperature were measured daily and maintained at 6.8 + 1.45 mg/L and 24 + 1.0 °C, respectively. The swimming and feeding behavior for all studied fish were observed. The current work was conducted according to the Egyptian laws and University guidelines for animal care and was approved by The National Ethical Committee of the Faculty of Veterinary Medicine, Assiut University, Egypt.

### 2.2. Histological and Histochemical Analysis

Fish were randomly collected from the tanks and euthanized with an overdose of MS-222 before tissue sampling. The spleen was removed, and excessive tissues were discarded. The specimens were rapidly dissected at 1 × 1 × 0.5 cm and were fixed in Bouin’s solution for 22 h. After proper fixation, specimens were processed through ethanol dehydration, methyl benzoate clearance, and paraffin embedding. Paraffin sections of 5 µm were obtained and stained with Harris hematoxylin and Eosin (HX), Periodic Acid-Schiff (PAS), Alcian blue (AB), Crossmon’s trichrome, and Grimilus Silver stains [17,18].

### 2.3. Semithin Sections and Transmission Electron Microscopy (TEM) Analysis

Small specimens of the spleen were immediately washed and fixed in a solution of 2.5% paraformaldehyde-glutaraldehyde overnight [19]. Samples were then washed in 0.1 Mol/L phosphate buffer and osmicated with 1% osmium tetroxide in 0.1 mol/L sodium cacodylate buffer at pH 7.3. Following that, the specimens were passed in ethanol followed by propylene oxide for dehydration and finally underwent Araldite embedding. Semithin sections of about one μm thickness were stained with toluidine blue and were examined under a light microscope. Ultrathin sections of about 70 nm were cut using Ultrotom-VRV (LKB, Bromma, Germany), and were stained with lead citrate and uranyl acetate [20]. TEM images were analyzed using a JEOL-100CX II electron microscope (Massachusetts, Boston, MA, USA).

### 2.4. Immunohistochemical Analysis

Immunohistochemical analysis of the spleen specimens was performed using a Pierce Peroxidase Detection Kit (36000, Thermo Fisher Scientific, Waltham, MA, USA). The sections were deparaffinized using xylene, rehydrated with a graded series of ethanol, washed with distilled water, and heated in a sodium citrate buffer (pH 6.0) for 15 min in a microwave to increase epitope exposure. After that, the sections were cooled for 30 min at room temperature, washed with a wash buffer (Tris-buffered saline with 0.05% Tween-20 Detergent), and were then incubated in a peroxidase suppressor for 30 min to quench the endogenous peroxidase activity. The sections were washed two times for 3 min with a wash buffer, blocked with a universal blocker™ blocking buffer in TBS for 30 at room temperature, and were incubated overnight at 4 °C with diluted (1:100) primary antibodies against nuclear factor kappa B (NF-κB), interleukin 1 beta (IL-1β), transforming growth factor-beta (TGf-β), autophagy protein 5 (APG5), and nuclear factor erythroid 2-related factor 2 (Nrf2) (sc-8008, sc-7884, sc-220, sc-133158, and sc-722, respectively, Santa Cruz Biotechnology, Heidelberg, Germany), in addition to myostatin and SRY-Box transcription factor 9 (Sox9) (AB3239 and AB5535, respectively, Sigma-Aldrich, Madrid, Spain). The sections were washed with a wash buffer for 3 min and were incubated with diluted (1:100) goat anti-mouse IgG (31800, Invitrogen, Waltham, MA, USA) and diluted (1:1000) goat anti-rabbit IgG (65-6140, Invitrogen, Waltham, MA, USA) secondary antibodies for 30 min at room temperature. After washing with a wash buffer, the tissues were incubated with the diluted (1:500) Avidin-HRP (43-4423, Invitrogen, Waltham, MA, USA) in a universal blocker blocking buffer for 30 min. The sections were then washed three times for 3 min each with a wash buffer and were incubated with 1× metal-enhanced DAB solution (by adding stable peroxide buffer to the 10× DAB/Metal Concentrate) for 5–15 min until the desired staining was achieved. Finally, the sections were washed with a wash buffer, counterstained with Harris modified hematoxylin, and mounted with mounting media.

### 2.5. Digitally Colored TEM Images

TEM images were digitally colored to enhance the visual contrast between the various structures present on the individual electron micrograph and to make these components more visible to the readers. All the structures were hand colored with the aid of Adobe Photoshop software version 6 (Adobe, San Jose, CA, USA).

## 3. Results

### 3.1. Histological and Histochemical Analysis

The spleen of the molly fish was a solitary peritoneal organ close to the liver (Figure 1A) and was characterized by well-distinct melanomacrophage centers. The main splenic structures of higher vertebrates were typically observed here in the spleen of the fish including red and white pulps, blood vessels, and ellipsoids. The red pulp consisted of an extensive network of splenic cords and sinusoidal capillaries (Figure 1B,C). The ellipsoids were thick-walled capillaries with a narrow lumen (Figure 1D) that resulted from the division of the splenic arterioles (Figure 1E). These were surrounded by a sheath of collagen and reticular fibers (Figure 1F).

Semithin sections showed that the white pulp was comprised of sheaths of leukocytes, consisting mainly of lymphoid cells (T lymphocytes) and macrophages with phagocytosed materials (Figure 2A,B). The germinal center was lacking. The red pulp fundamentally consisted of erythroid cells and thrombocytes, which comprised most of the splenic parenchyma (Figure 2A,B). Splenic cords were a tangle of fibroblast-like cells with foci of various blood cells that typically surrounded arterial vessels and ellipsoids (Figure 2C). Some mitotic cells could be identified in the red pulp (Figure 2C). The ellipsoids consisted of cuboidal endothelial cells. They passed via a reticular cell sheath, and macrophages (Figure 2D). Melanomacrophage centers (MMCs) were immune cell types formed of aggregations of macrophages and contained varying amounts of pigment, localized in vacuoles (Figure 2E,F).

The semithin sections showed sporadic neurons with fine cell processes that appeared pale in color with a vesicular nucleus surrounded by metachromatic Nissl granules (Figure 3A,B). Mast cells with characteristic metachromatic granules could be seen among the splenic cells (Figure 3C). Rodlet cells were also observed around the ellipsoids that were characterized by thick capsules and many cytoplasmic rodlet granules (inclusions) (Figure 3C,D). Many branched epithelial reticular cells and monocytes with eccentric kidney-shaped nuclei could be identified in the spleen (Figure 3E,F).

The spleen was innervated by autonomic sympathetic fibers (splanchnic nerves) (Figure 4A,C) that showed positive reactions to silver impregnation. Moreover, adrenergic neurons were distributed singularly in the splenic parenchyma and showed positive reactions to silver impregnation (Figure 4B). Rodlet cells were scattered in the spleen and showed argyrophilic properties (Figure 4C–E).

### 3.2. Immunohistochemistry

The white pulp of the spleen expressed APG5 (Figure 5A). The expressions were well distinct in the melanomacrophages (Figure 5B), leukocytes (Figure 5C), and macrophages (Figure 5D). Myostatin was expressed in leukocytes and epithelial reticular cells (Figure 5E,F). IL-1β showed immunoreactivity in monocytes and macrophages around the ellipsoid (Figure 6A,B). NF-κB was expressed in macrophages and epithelial reticular cells (Figure 6C,D). TGF-β was also expressed in the macrophages and reticular cells (Figure 6E,F). Nrf2 was expressed in stem cells (Figure 7A) and rodlet cells (Figure 7B). Sox-9 was expressed in the epithelial reticular cells (Figure 7C) and stem cells (Figure 7D).

### 3.3. Electron Microscopy

The splenic parenchyma was comprised mainly of red pulp intermingled with clusters of white pulp cells (Figure 8A). The main cells identified in the spleen of molly fish included:

Erythrocytes: They were widely distributed in the red pulp and appeared as oval cells with large oval nuclei (Figure 8A);

Neurons: Large cell with cell processes containing a large euchromatic nucleus, surrounded by rER, lysosomes, and mitochondria (Figure 8B);

Dendritic cells: They exhibited indented heterochromatic nuclei, pseudopodia, and cytoplasmic lysosomes (Figure 8C);

Stem cells: They were characterized by nuclear division and a higher nuclear to cytoplasmic ratio (Figure 8C);

Epithelial reticular cells: These cells showed a branched appearance (Figure 8C), in which their processes extended between the lymphocytes and erythrocytes (Figure 9A). They showed a connection with macrophages and their cytoplasm possessed numerous ribosomes, and electron-lucent vesicles (Figure 9B). Moreover, they resembled the pericytes or the cells lining the blood vessels similar to those demonstrated in the spleen and bone marrow of mammals. These cells were characterized by the existence of mitochondria and rER (Figure 9C,D);

Lymphocytes: They were one of the most predominant cell types in the spleen. They displayed a higher nuclear to cytoplasmic ratio and heterochromatic nuclei (Figure 9 and Figure 10A–D);

Ellipsoids: The ellipsoids were lined with a simple cuboidal epithelium with a central irregular nucleus with heterochromatin clumps. The cytoplasm contained lysosomes, mitochondria, and vacuoles (Figure 10A);

Granular leukocytes that included:

Eosinophils: (Figure 10A,B) displayed pseudopodia and an eccentrically located horseshoe-shaped nucleus. The cytoplasm displayed rounded electron-dense granules and vesicles;

Basophils: The cytoplasm of basophils was packed with electron-dense granules (Figure 10C). Furthermore, they showed pseudopodia and the nucleus displayed segmentation;

Neutrophils: (Figure 10D) were frequently observed in the spleen of molly fish and were characterized by a segmented nucleus with dense and patchy distributed nuclear chromatin. The cytoplasm demonstrated many electron-dense granules, rER, vacuoles, and some phagocytosed materials;

Macrophages: Macrophages were the most common cells found in the spleen. Their plasma membrane exhibited long pseudopodia and the cytoplasm possessed many lysosomes and other phagocytosed materials (Figure 11A,B). They were usually associated with dendritic cells and leukocytes. The active macrophages displayed irregular shapes and were characterized by the kidney-shaped eccentric euchromatic nucleus (Figure 11C). They were integrated to form melanomacrophage centers (MMCs), with vesicles containing pigments (Figure 11D).

## 4. Discussion

The immune system is classified into two immune systems: innate and adaptive. It has a series of physical barriers, in addition to humoral and cellular components that are responsible for the body’s defense against pathogens and foreign substances [13,21]. The innate immune system is the main defense in invertebrates, and these animals cannot produce humoral or cellular immune responses [22]. In teleosts, the immune system includes the thymus, spleen, head kidney, and the mucosa-associated lymphoid tissues (MALT) [23,24]. In fish, the spleen is the major secondary lymphoid organ. It contains a significant number of (IgM + B) lymphocytes in addition to its role in the induction of adaptive immune responses. Moreover, it is essential for the elimination of immune complexes [25].

The spleen of the molly fish was characterized by the presence of well-distinct melanomacrophage centers (MMCs) that were formed of macrophage aggregations, with varying amounts of pigment, localized in vacuoles. These macrophages were usually associated with dendritic cells and leukocytes. The MM is a distinctive and predominant immune cell type in the spleen of teleosts. These cells contain varying amounts of pigment localized in vacuoles such as melanin (black–brown), ceroid, hemosiderin, or lipofuscin (yellow to golden brown) localized in vacuoles. The MMCs are believed to have a scavenger structure; however, their role in the immune system is obscure. The size and number of MMCs revealed an increment with fish age [26]. Recently, MMCs have been reported to be an indicator of immune function in fish [27]. Furthermore, splenic MMCs or macrophage aggregates are considered one of the potential biomonitoring tools for determining the impacts of minute concentrations of pesticide contaminants [28,29,30,31]. Bols et al. [32] concluded that the increase in the number of MMCs could serve as a biomarker for toxic effects.

In fish, larger macrophage aggregations or MMCs are normally found in the lymphatic system and other organs, and it has been suggested that they are involved in various processes of storage, phagocytosis, and the detoxification of bacteria [33]. The number and size of MMCs and their pigment content varies depending on fish health and environmental degradation [34]; it has been suggested that increased numbers of MMCs in the fish spleen are induced by environmental factors and stress, rather than by tissue catabolism [35,36]. In fish, the accumulation of melanin in the macrophages plays a critical function in the neutralization of toxic free radicals which are produced during the process of unsaturated lipid peroxidation [37]. Genes involved in the melanogenesis pathway are expressed in the spleen, suggesting melanin synthesis in the organ [38,39].

The spleen of the molly fish demonstrated the basic elements of the higher vertebrates, where the spleen depicted blood vessels, red and white pulps, in addition to ellipsoids that result from the splitting of the splenic arterioles. A few species of fish are characterized by the lack of these ellipsoids such as Anguilliformes [40]. It has been suggested that ellipsoids have an expert function in plasma filtration. Moreover, foreign bodies, such as bacteria, are restricted by the ellipsoids and may be found within the reticular meshwork or within the macrophages of the sheaths intracellularly [41].

The white pulp of the spleen in the molly fish was comprised of sheaths of leukocytes, consisting mainly of lymphoid cells (T lymphocytes) and macrophages. The white pulp has an organized lymphoid structure [42]. T lymphocyte sub-populations are a characteristic feature of the immune system of teleosts. These cells are very important for the adaptive immune system [43]. The granular leukocytic cells detected in the spleen here were eosinophils, basophils, neutrophils, and macrophages. Kondera [44] reported that the most frequent leucocytic cells demonstrated in the hemopoietic tissues are lymphocytes, neutrophils, and macrophages. These cellular elements are the master cells implicated in phagocytosis in fish and are responsible for the removal of bacteria through the production of reactive oxygen species (ROS) during a respiratory burst [45].

Rodlet cells were randomly distributed in the spleen and were also observed around the ellipsoids. Rodlet cells have been observed in various fish organs including the thymus, spleen, kidney, heart, gills, gall bladder, pancreas, liver, skin, and blood vessels [46]. Previous studies have proposed the association of these cells as a part of a normal population with the fish defense system [47]. Rodlet cells may be embroiled in water or electrolyte transport, pH control, and lubrication, in addition to their antibiotic effects, and may also be deemed as non-specific immune cells, implicated in immunity, where the number of rodlet cells is boosted in parasitic infection [48].

Dendritic cells were detected among the splenic cells. These cells are one of the antigen-presenting cells with phagocytic ability, dendritic morphology, motility, and T cell-catalyzed properties [49]. Functional dendritic cells have been recognized in different tissues of some teleosts, including the spleen [50]. These cells are important for providing the indispensable interactions for the function of endothelial and lymphoid cells of blood vessels [51].

The white pulp of the spleen expressed APG5, and the expressions were well distinct in the melanomacrophages, leukocytes, and macrophages. APG5 is one of the essential players in the autophagy process [52], and it is critical for multiple processes including autophagic vesicle formation, lymphocyte development and proliferation, mitochondrial quality control, and apoptosis [53].

Myostatin was expressed in the leukocytes and epithelial reticular cells of the molly fish spleen. The myostatin precursor was detected in several teleost fish tissues including the spleen [54]. It is a differentiation and growth factor of the TGF-β superfamily, and it inhibits skeletal muscle development and growth [55,56]. Myostatin expression was found in mouse spleen, supposing the possible effects of myostatin on immune cell development in mammals and fish [57].

IL-1β immunoreactivities were detected in monocytes and macrophages around the ellipsoid. It is one of the earliest expressed pro-inflammatory cytokines, which enable organisms to respond immediately to infection [58]. IL-1β was significantly expressed in many tissues of the pufferfish including the spleen. It is a critical pro-inflammatory cytokine, which mediates the regulation of innate and adaptive immune responses. IL-1β was reported to be secreted by activated endothelial cells, tissue macrophages, blood monocytes, activated T lymphocytes, granulocytes, and other cell types [59]. It affects various types of cells and plays a major role in the launch of local and systemic responses to various stimuli such as infection or injury through T and B lymphocytes, activating macrophages, and natural killer cells [60,61]. It has been proposed that IL-1β is released from cells by a non-classical secretory pathway [62].

In fish, the production of recombinant IL-1β in either the precursor or mature form to implement functional analyses has been described in several studies [63,64,65]. These studies confirmed that fish IL-1β proteins have antibacterial and antiviral defense activities, where these proteins are involved in the regulatory mechanisms of the inflammatory response to bacterial and/or parasitic infection [66,67]. Thus, IL1-β is an essential mediator of early infection responses and a critical cytokine linked to inflammation [68].

The spleen of the molly fish revealed expressions of NF-κB and TGF-β in the macrophages and epithelial reticular cells. NF-κB responds to inflammatory and immune stimuli and regulates cell proliferation, adhesion, invasion, apoptosis, and angiogenesis in multiple cell types [69]. Furthermore, its signals within epithelial cells play an essential role in maintaining immune homeostasis in barrier tissues [70]. NF-κB is a critical transcription factor in the innate immune response. It interposes the production of many pro-inflammatory cytokines and has an essential role in numerous signaling pathways [71,72].

TGF-β is a pleiotropic cytokine produced by various cells including immune cells and non-hematopoietic cells. It has significant impacts on cell proliferation, oncogenesis, and immune response suppression, in addition to the suppression of intestinal inflammatory responses to bacterial antigens [73]. TGF-β is released from fibroblasts, platelets, and macrophages at first [74], and has an essential immunoregulatory role in mammalian innate and adaptive immune pathways [75]. Moreover, TGF-β regulates the active and inactive states of macrophages and monocytes under specific conditions [76]. Higher expression of TGF-β1 was detected in immune-associated tissues of fish, including the spleen, thymus, and head kidney [77].

Nrf2 was expressed in stem cells and rodlet cells within the spleen of molly fish. Nrf2 has been reported to be involved in antioxidation, immunopotentiation, and osmoregulation in fish under salinity stress, in addition to its role in toxicity and oxidative stress [78,79]. Nrf2 has significant roles in the maintenance of the intestinal mucosal barrier integrity, where it minimizes intestinal mucosal injury and inflammation, regulates intestinal permeability, and influences the differentiation and function of T cells [80].

Interestingly, the spleen of the molly fish showed expressions of Sox9 in epithelial reticular cells and stem cells. Sox9 is a member of the SOX family, which participates in the regulation of cell proliferation and cell fate during embryogenesis [81,82], and its mutations result in abnormal cell growth [83]. The Sox family have critical roles in stem cell maintenance, embryonic development, and lineage commitment [84], where Sox9 regulates stem and progenitor cells in adult tissues [85].

## 5. Conclusions

The current study demonstrated a variety of immune cell types with specific functions in the spleen of molly fish according to their morphological characteristics and immunohistochemical expressions. The strong positive immunoreactivity of various splenic components and their inflammatory cells suggests its critical role in immunity. The results of the current study may be summarized as follows: (i) the IL-1β immunoreactivity in monocytes and macrophages suggests the role of the spleen in the regulation of both adaptive and innate immunity, cell proliferation, and apoptosis; (ii) the NF-κB and TGF-β expression in macrophages and epithelial reticular cells suggests the spleen has a role in immune response suppression and cell proliferation regulation, besides its role in maintaining immune homeostasis; (iii) myostatin immunoreactivity in leukocytes and epithelial reticular cells suggests the role of the spleen in immune cell development; (iv) the Nrf2 immunoexpression in rodlet cells and stem cells suggests the role of the spleen in immunopotentiation, in addition to its function in toxicity and oxidative stress; (v) the APG5 expression in the melanomacrophages, leukocytes, and macrophages suggests the role of the spleen in lymphocyte development and proliferation; (vi) the Sox9 immunoreactivity in epithelial reticular cells and stem cells suggests the role of the spleen in cell proliferation and regulation and stem cell maintenance.

## Figures and Tables

**Figure 1 biology-11-00779-f001:**
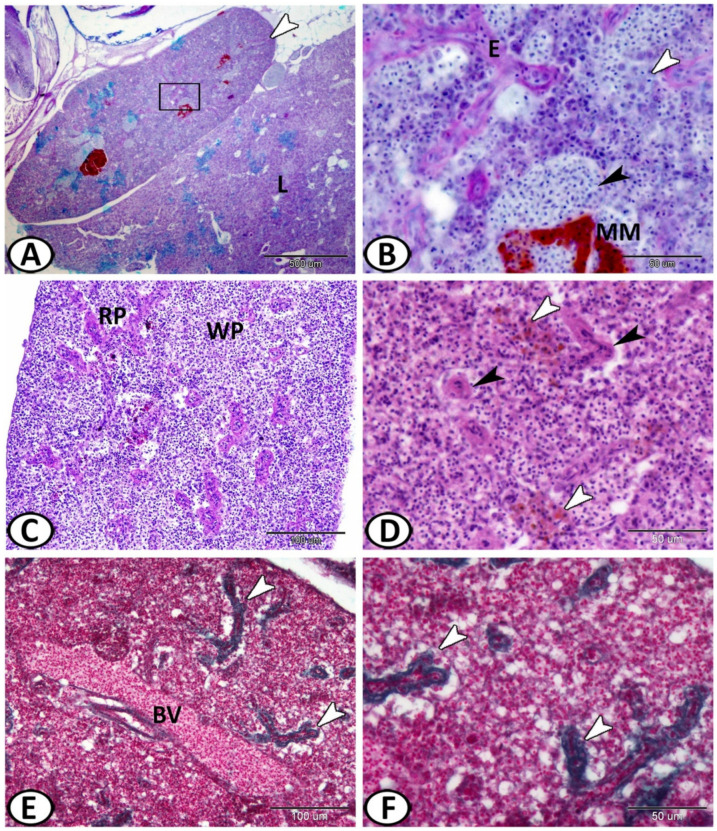
(**A**) General view of the spleen (white arrowhead) of molly fish found adjacent to the liver (L) and showing the distribution of melanomacrophage centers (red color) (AB/HX). (**B**) Higher magnification of the boxed area shows white pulp (white arrowhead), the red pulp (black arrowhead), ellipsoids (E), and melanomacrophage center (MM) (AB/HX). (**C**) The parenchyma of the spleen of Molly fish consists of white pulp (WP) and red pulp (RP) (PAS/HX). (**D**) Ellipsoids (black arrowheads) and melanomacrophage center (white arrowheads) (HE). (**E**,**F**) Ellipsoids (white arrowheads) are the termination of splenic arterioles. Note a large blood vessel (BV) (Crossmon’s trichrome).

**Figure 2 biology-11-00779-f002:**
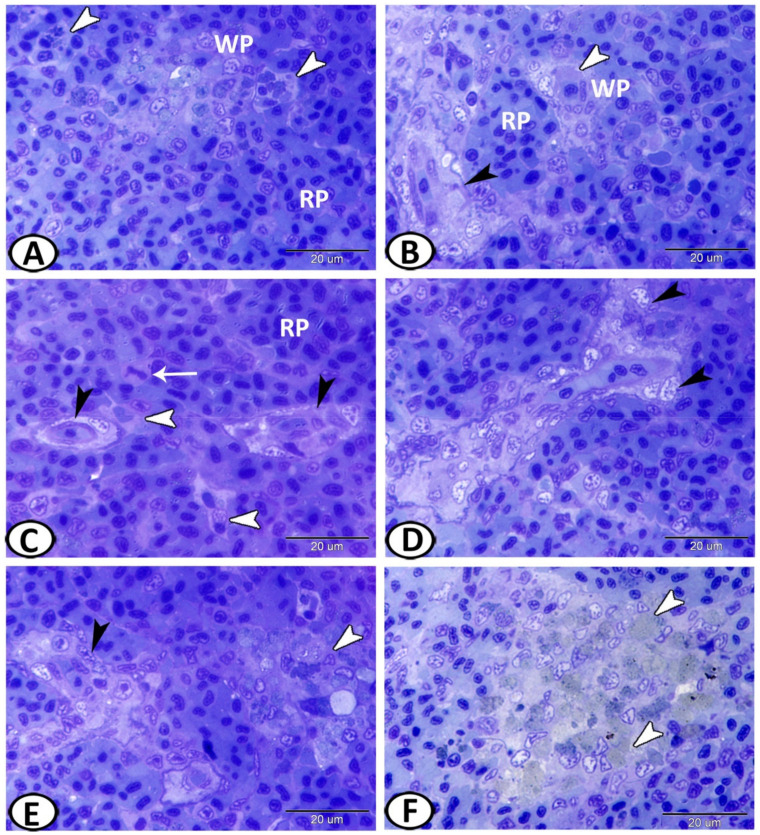
Semithin sections of the spleen stained with toluidine blue. (**A**,**B**) The spleen of molly fish illustrates the components of the parenchyma, the red pulp (RP), and the white pulp (WP). Note the macrophages (white arrowheads) and ellipsoids (black arrowheads). (**C**) The red pulp (RP) consisted mainly of erythroid cells, which comprised the majority of the splenic parenchyma. Splenic cords were a mesh of fibroblast-like cells (black arrowheads) that typically surrounded ellipsoids. Note the mitotic cells (arrow) and macrophages (white arrowheads). (**D**) The ellipsoids consisted of cuboidal endothelial cells (black arrowheads). (**E**,**F**) Melanomacrophage centers (white arrowheads) contained varying amounts of pigment localized in close relation to the ellipsoids (black arrowheads).

**Figure 3 biology-11-00779-f003:**
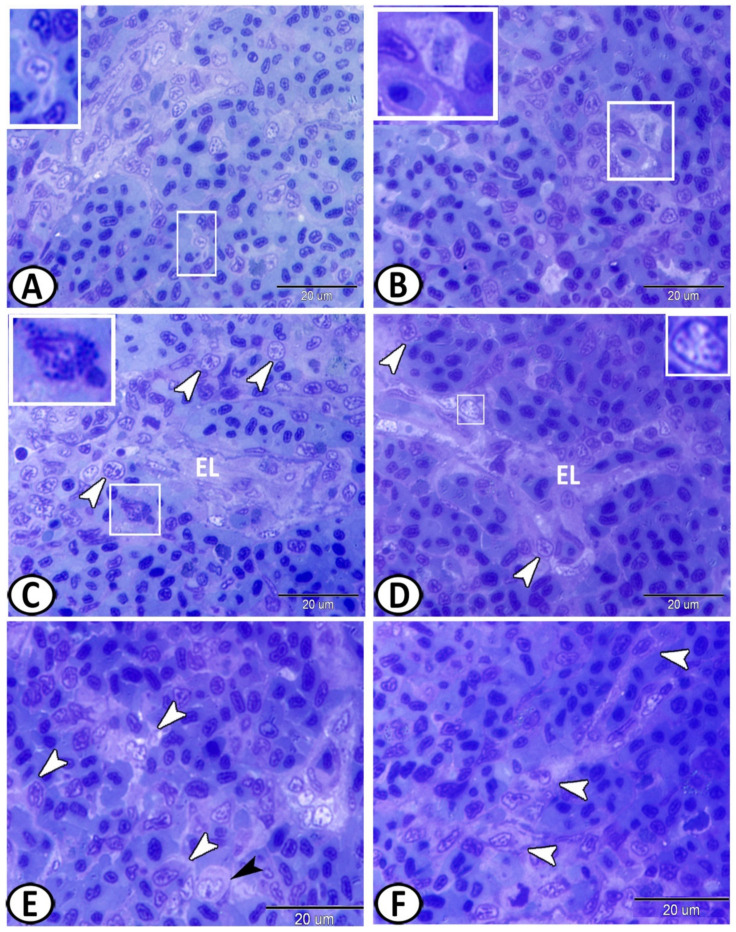
Semithin sections of the spleen stained with toluidine blue. (**A**,**B**) neurons (boxed areas) were observed in the splenic parenchyma. (**C**) Mast cells with characteristic metachromatic granules (boxed areas) and rodlet cells (arrowheads) were present among the splenic cells. (**D**) Rodlet cells (arrowheads, boxed areas) were distributed around the ellipsoids (EL). (**E**,**F**) Many branched epithelial reticular cells (white arrowheads) and monocytes (black arrowheads) were identified in the spleen.

**Figure 4 biology-11-00779-f004:**
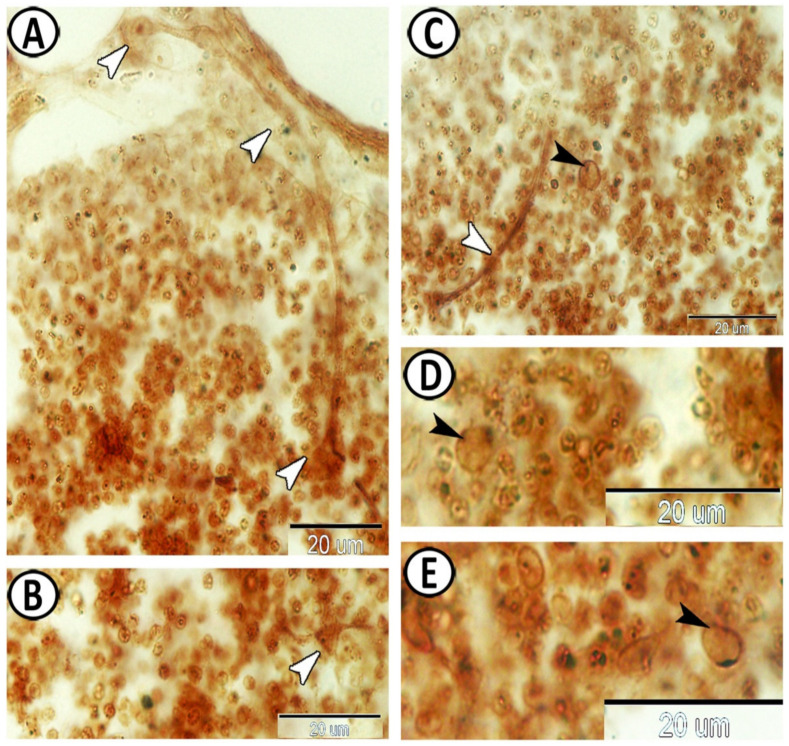
The spleen of molly fish stained with using the Grimilus Silver method. (**A**) The spleen was innervated by sympathetic nerve fibers (arrowheads). (**B**) A single adrenergic neuron (arrowhead) was found in the splenic parenchyma. (**C**–**E**) Rodlet cells (black arrowheads) were randomly distributed in the spleen. Note the nerve fiber (white arrowhead) among the splenic cells.

**Figure 5 biology-11-00779-f005:**
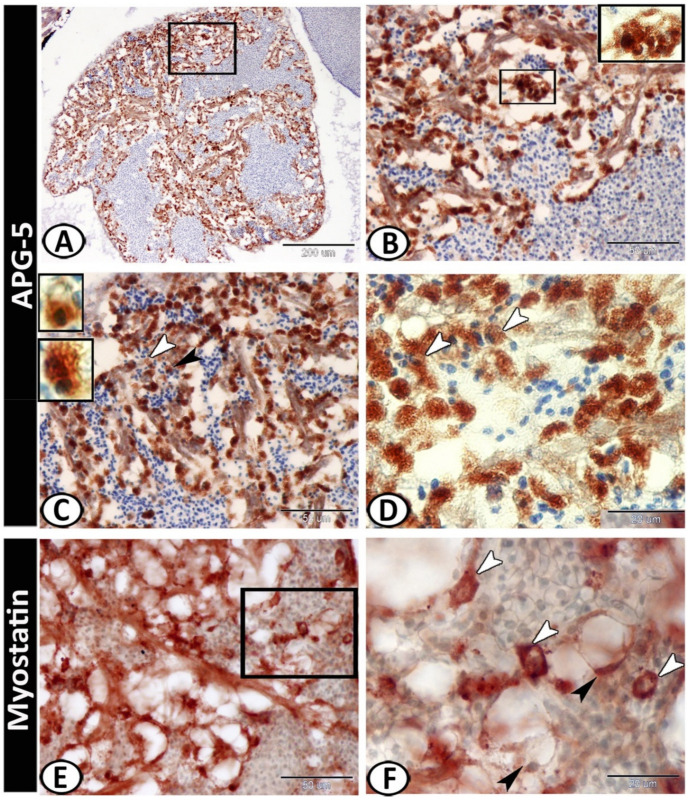
Immunohistochemistry of APG5 and Myostatin in the spleen. (**A**) The white pulp of the spleen expressed APG5 (boxed area). (**B**) APG5 expressions were well distinct in the melanomacrophages (boxed areas). (**C**) Leukocytes (boxed areas, arrowheads) expressed APG5. (**D**) Macrophages (arrowheads) also expressed APG5. (**E**,**F**) Myostatin was expressed in the leukocytes (white arrowheads) and epithelial reticular cells (black arrowheads).

**Figure 6 biology-11-00779-f006:**
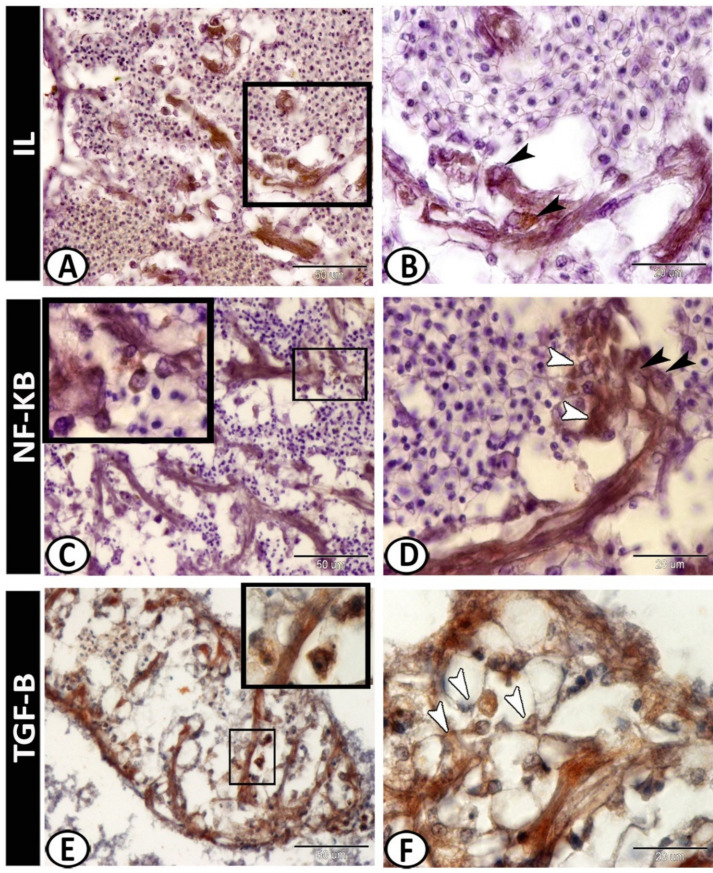
Immunohistochemistry of IL-1β, NF-κB, and TGF-β in the spleen. (**A**,**B**) IL-1β showed immunoreactivity in monocytes and macrophages around the ellipsoids (arrowheads). (**C**,**D**) NF-κB was expressed in the macrophages (boxed areas, white arrowheads) and epithelial reticular cells (black arrowheads). (**E**,**F**) TGF-β was expressed in the macrophages (boxed areas) and reticular cells (arrowheads).

**Figure 7 biology-11-00779-f007:**
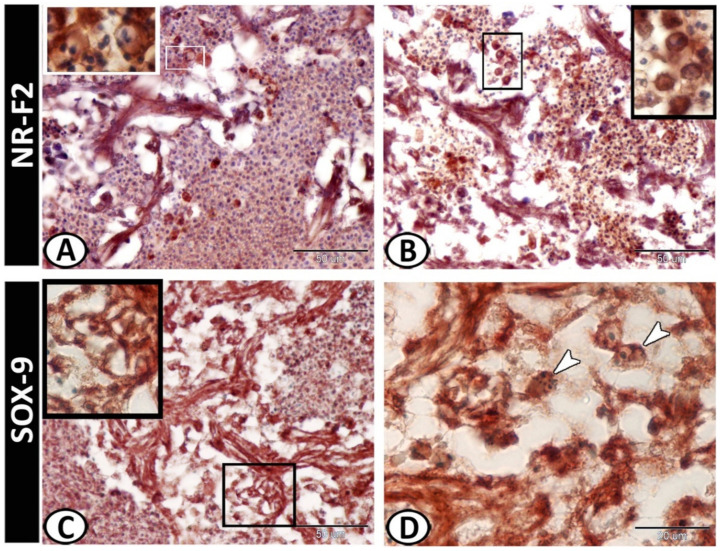
Immunohistochemistry of NR-F2 and Sox-9 in the spleen. (**A**) NR-F2 was expressed in stem cells (boxed areas). (**B**) NR-F2 was expressed in the rodlet cells (boxed areas). (**C**) Sox-9 was expressed in epithelial reticular cells (boxed areas). (**D**) Sox-9 showed immunoreactivity in the stem cells (arrowheads).

**Figure 8 biology-11-00779-f008:**
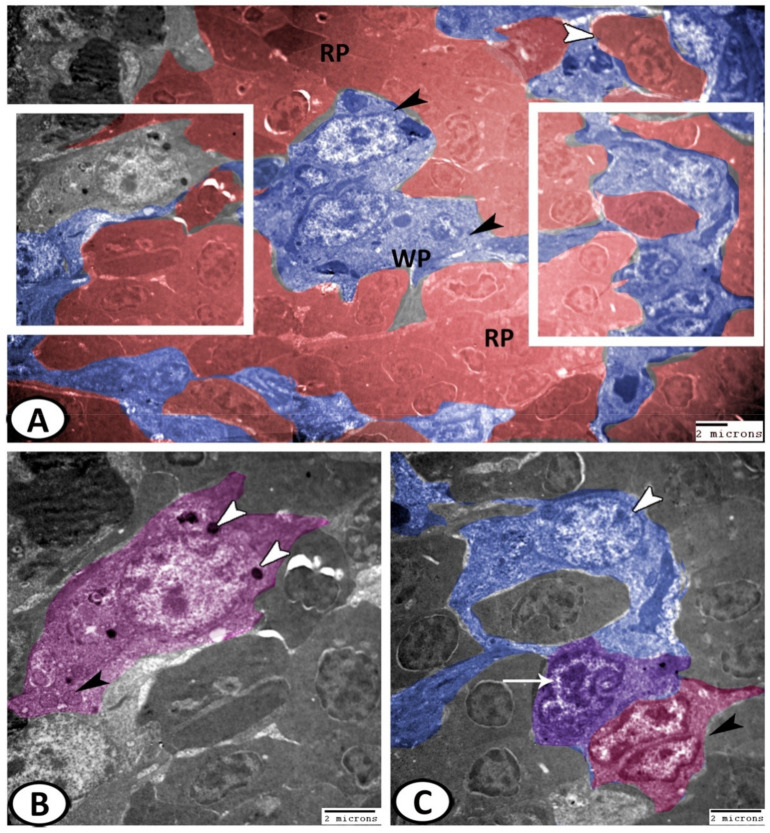
Digital colored TEM of the cells in the spleen. (**A**) The splenic parenchyma was comprised mainly of the red pulp (red, RP) intermingled with clusters of white pulp cells (blue, WP). Note, erythrocytes (white arrowheads) and macrophages (black arrowheads) with phagocytosed materials. (**B**) Higher magnification of the left boxed area in (**A**) illustrating neuron (pink) that contained a large euchromatic nucleus, mitochondria (black arrowhead), and lysosomes (white arrowheads). (**C**) Higher magnification of the right boxed area in (**A**) showing that dendritic cells (violet, arrow) with an indented heterochromatic nucleus and stem cells (pink, black arrowhead) with a dividing nucleus were observed in the splenic parenchyma. Note the branched epithelial reticular cells (blue, white arrowhead).

**Figure 9 biology-11-00779-f009:**
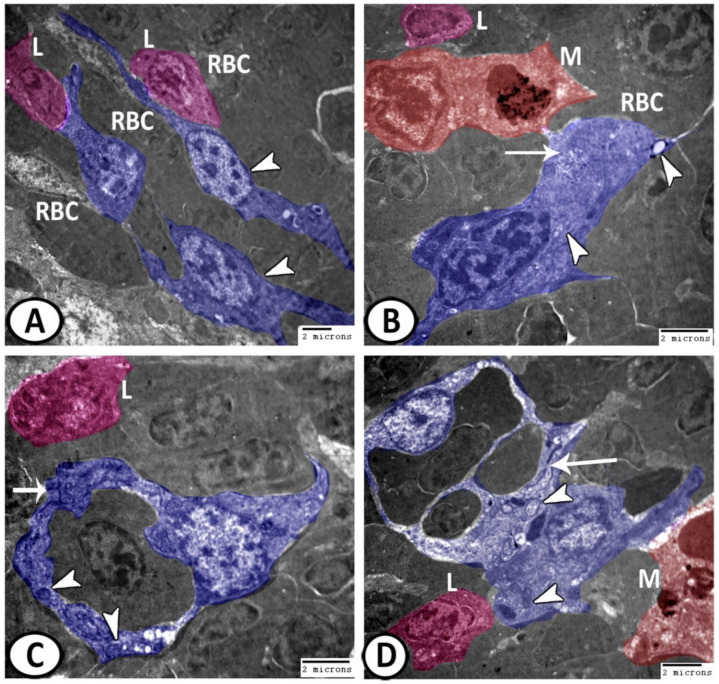
Digital colored TEM of the epithelial reticular cells in the spleen. (**A**) The epithelial reticular cells (blue, arrowheads) extended their processes between the lymphocytes (L, pink) and erythrocytes (RBC). (**B**) They showed a connection with macrophages (M, red) and their cytoplasm possessed numerous ribosomes (arrow) and electron-lucent vesicles (white arrowheads). (**C**,**D**) They were characterized by the presence of mitochondria (arrowheads) and rER (arrows). Note the presence of macrophages (M, red) and lymphocytes (L, pink).

**Figure 10 biology-11-00779-f010:**
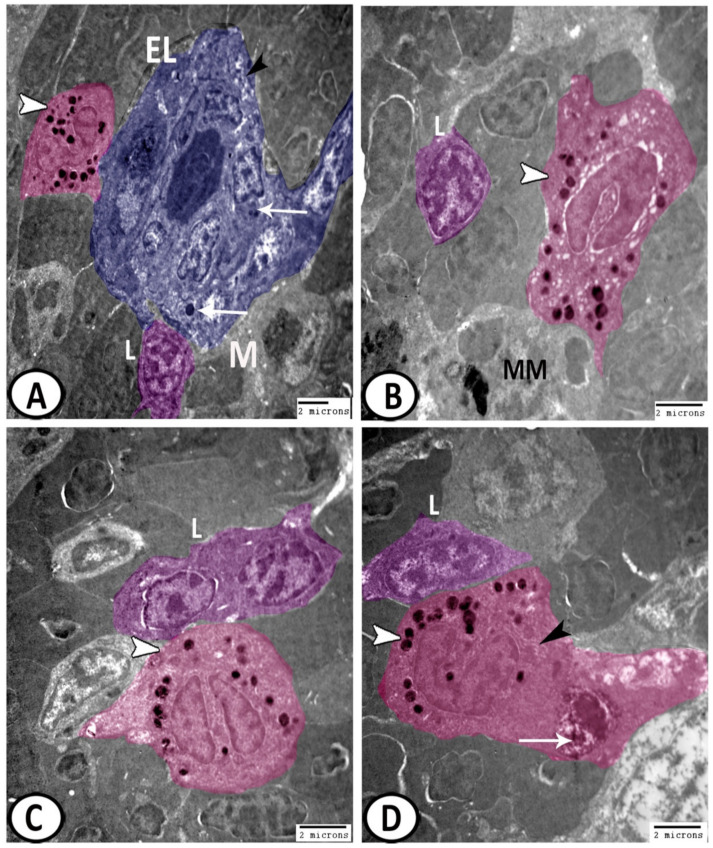
Digital colored TEM of the leukocytes in the spleen. (**A**) The ellipsoids (blue, E) were lined with a simple cuboidal epithelium and contained lysosomes (arrows), and vacuoles (black arrowhead). Note the surrounding macrophage (M), lymphocytes (L, violet), and eosinophils (pink, white arrowhead). (**B**) Eosinophils (pink, arrowhead) displayed horseshoe-shaped nuclei and rounded electron-dense granules and vesicles. Note the presence of lymphocytes (L, violet) and melanomacrophages (MM). (**C**) The cytoplasm of basophil (pink, arrowhead) was packed with electron-dense granules. Note the associated lymphocytes (L, violet). (**D**) Neutrophils (pink) with a segmented nucleus, and many electron-dense granules (white arrowhead), rER (black arrowhead), and phagocytosed materials (arrow). Note the presence of lymphocytes (L, violet).

**Figure 11 biology-11-00779-f011:**
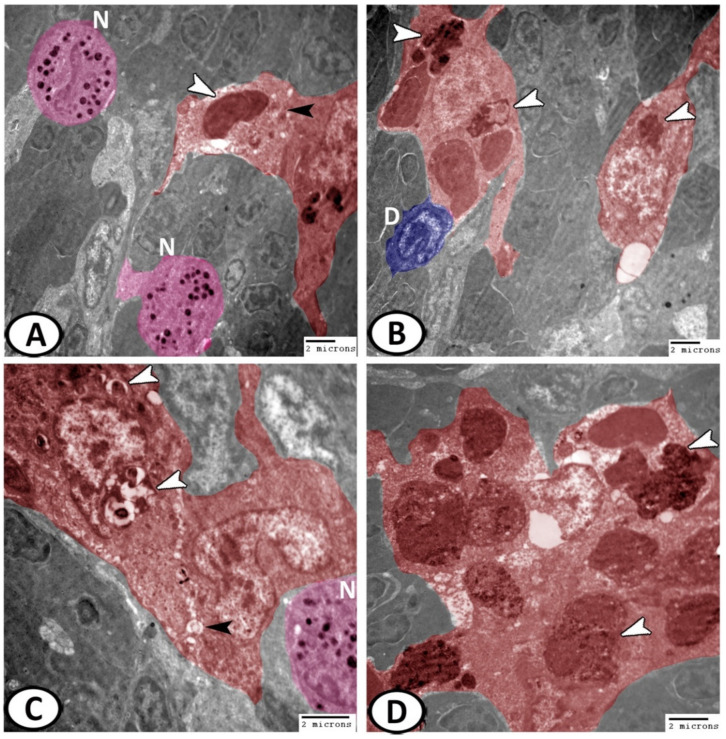
Digital colored TEM of the macrophages in the spleen. (**A**,**B**) Macrophages (red) displayed long pseudopodia and their cytoplasm contained many lysosomes (black arrowheads) and other phagocytosed materials (white arrowheads). They were usually associated with dendritic cells (blue, D) and neutrophils (pink, N). (**C**) The active phagocytosing macrophages (red) were characterized by the kidney-shaped nucleus, vacuoles (black arrowheads), and phagocytosed materials (white arrowheads). Note the presence of neutrophils (pink, N). (**D**) They were merged into spherical structures known as melanomacrophage centers (MMCs, red) with vesicles containing pigments (arrowheads).

## Data Availability

The data presented in this study are available within the article.

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
