# Peer review of "Structural and Functional Aspects of the Spleen in Molly Fish Poecilia sphenops (Valenciennes, 1846): Synergistic Interactions of Stem Cells, Neurons, and Immune Cells"

_biology, 2022, doi:10.3390/biology11050779_

Round 1
Reviewer 1 Report
The Authors of the reviewed manuscript titled "Structural and functional aspects of the spleen in molly fish Poecilia sphenops (Valenciennes, 1846): Synergistic interactions of stem cells, neurons, and immune cells" have prepared a solid histological description of different cell types found within the spleen of the molly, a popular species of livebearing ornamental fish. Generally, this paper deserves to be published, as the topic of fish spleens is underappreciated and any new piece of information is valuable, but it requires some corrections to be made in the text, mostly within the non-technical parts.
Especially, the text needs to be checked in terms of the quality of English language, as it is noticeably lacking in this regard. I highlighted only the most jarring mistakes, meanwhile, the overall style in most of the other sentences can also be improved (most noticeably in the Simple Summary, Abstract and Introduction, but also partially in the Discussion).
I have aligned my commentary in a paragraph by paragraph manner, to be seen below. I also attached an edited .pdf file with the same comments and markings, for convenience.
Simple Summary: This paragraph requires some linguistic corrections, as it is quite "clunky" in its current form.
- Lines 18-20: Merge these two sentences into one and improve the style/phrasing.
- Line 20: Delete "regarded as being".
- Line 24: Change "was" to "is" and "correct to "white pulp".
- Line 27: Move the closing parenthesis after "neutrophils".
- Lines 28-29: Please improve the style/phrasing of this last sentence.
Abstract: Same as above - some sentences need to be corrected, as they affect the overall perception of this paragraph. I would also suggest to add a sentence which will contain information about the number and size of studied fish, as well as the used scientific methodology.
- Lines 30-31: The opening sentence must be improved in terms of style/vocabulary.
- Lines 31-32: Change the word order to "This study used the molly fish (Poecilia sphenops) as a model ...".
- Line 41: Correct to plural "ellipsoids".
- Lines 43-44: This final sentence needs to be replaced, it is far too simplistic and it merely repeats a very well known and basic fact about the importance of the spleen for fish immunity. Focus more on what was actually proven here and what are the outcomes for future research.
Introduction: Apart from linguistic issues, I suggest to expand the description of the molly fish - why is it such a popular aquarium species? What is so interesting about its reproductive biology? This way, the actual point of studying the spleen of this exact species will be emphasized more clearly.
- Line 56: Delete "of the organisms".
- Lines 58-59: Please improve the style/phrasing of the second part of this sentence.
- Lines 62-63: Delete "where this fish category lacks lymph nodes and bone marrow,".
- Lines 60-65: Actually, I believe this whole paragraph is unnecessary and can be deleted.
- Line 77: I believe it should be "floating" instead of "fluctuating".
- Lines 80-82: What is "expected"? This first half of the final sentence is barely understandable, please rewrite this part.
Materials and Methods: Apart from a few minor issues, this section contains the necessary information, especially the description of IHC seems to be very detailed.
- Line 87: Correct to singular "Sample".
- Line 90: Correct the unit to "g".
- Line 96: How were the fish euthanized?
- Line 115: Change "were" to singular "was".
- Line 121: Change "conquer" to "quench".
- Lines 125-131: I believe there is no need to indicate the full information about the producers repeatedly in each parenthesis. Instead, it may be grouped in several ways.
- Line 145: Change "as a trial" to "in an attempt".
Results: The quality of the presented Figure panels is on a satisfactory level. However, I spotted a few ambiguities within the written part of this section, so please pay attention to these details during the review.
- Line 156: What is "that open in the blood vessels" supposed to mean? There must be a mistake. Please correct.
- Line 168: Again, the ellipsoids "open to blood vessels"?
- Line 184: Delete "(Fig. 2A, B)".
- Line 185: Delete "(Fig. 2C)" and correct to "that typically surrounded ellipsoids".
- Line 193: "Capsules" of rodlet cells? What is that?
- Line 206: Correct to "singularly".
- Line 246: Correct to "pulp".
- Line 254: A pity that the Authors did not study a proliferation marker using IHC, such as PCNA. This would allow for a more firm identification of cellular proliferation.
- Line 260: I believe "cells" is missing in front of "lining".
- Line 291: Correct to "Fig. A".
- Line 308: Correct to plural "nuclei".
- Lines 322-323: Correct word order to "vesicles containing pigments".
Discussion: Overall, all findings described in the "Results" section were commented upon, what is definitely a good information. Furthermore, the English language in this section seems to be on a higher level than in other parts of the manuscript, but I still advise a full linguistic review of the entirety of the text.
- Line 325: Delete the starting sentence.
- Line 329: Change "this category" to "these animals".
- Line 348: "Increased trend"? What trend? Please rephrase.
- Line 362: Again, what does "open in the blood vessels" mean in the context of ellipsoids?
- Line 363: "A few species do not have ellipsoids" - what species? Citation?
Conclusions: Well written, comprehensive, no objections here. Well done.

Author Response
Reviewer 1
The Authors of the reviewed manuscript titled "Structural and functional aspects of the spleen in molly fish Poecilia sphenops (Valenciennes, 1846): Synergistic interactions of stem cells, neurons, and immune cells" have prepared a solid histological description of different cell types found within the spleen of the molly, a popular species of live bearing ornamental fish. Generally, this paper deserves to be published, as the topic of fish spleens is underappreciated and any new piece of information is valuable, but it requires some corrections to be made in the text, mostly within the non-technical parts.
Especially, the text needs to be checked in terms of the quality of English language, as it is noticeably lacking in this regard. I highlighted only the most jarring mistakes, meanwhile, the overall style in most of the other sentences can also be improved (most noticeably in the Simple Summary, Abstract and Introduction, but also partially in the Discussion).
We have carefully revised the entire manuscript considering this suggestion, with the support of a native English speaker.
I have aligned my commentary in a paragraph by paragraph manner, to be seen below. I also attached an edited .pdf file with the same comments and markings, for convenience.
We thank the Reviewer for the time spent carefully reviewing our manuscript. We have considered all your important comments and corrected them in the new version of the manuscript with all the requested changes. Thank you for your help. The Authors
Simple Summary: This paragraph requires some linguistic corrections, as it is quite "clunky" in its current form.
We corrected all the requested changes.
- Lines 18-20: Merge these two sentences into one and improve the style/phrasing.
Done
- Line 20: Delete "regarded as being".
We deleted it
- Line 24: Change "was" to "is" and "correct to "white pulp".
Corrected
- Line 27: Move the closing parenthesis after "neutrophils".
Done
- Lines 28-29: Please improve the style/phrasing of this last sentence.
We improved it.
Abstract: Same as above - some sentences need to be corrected, as they affect the overall perception of this paragraph. I would also suggest to add a sentence which will contain information about the number and size of studied fish, as well as the used scientific methodology.
We improved it and added information about the number of fish and methodology.
- Lines 30-31: The opening sentence must be improved in terms of style/vocabulary.
We improved it.
- Lines 31-32: Change the word order to "This study used the molly fish (Poecilia sphenops) as a model ...".
We improved it.
- Line 41: Correct to plural "ellipsoids".
Corrected
- Lines 43-44: This final sentence needs to be replaced, it is far too simplistic and it merely repeats a very well known and basic fact about the importance of the spleen for fish immunity. Focus more on what was actually proven here and what are the outcomes for future research.
We changed it.
Introduction: Apart from linguistic issues, I suggest to expand the description of the molly fish - why is it such a popular aquarium species? What is so interesting about its reproductive biology? This way, the actual point of studying the spleen of this exact species will be emphasized more clearly.
The entire manuscript was revised for linguistic issues. More information and features about molly fish were added to the last part of the introduction section in the new version of the manuscript.
- Line 56: Delete "of the organisms".
Done
- Lines 58-59: Please improve the style/phrasing of the second part of this sentence.
We improved it.
- Lines 62-63: Delete "where this fish category lacks lymph nodes and bone marrow,".
Done
- Lines 60-65: Actually, I believe this whole paragraph is unnecessary and can be deleted.
As suggested, we deleted the whole paragraph. Thank you.
- Line 77: I believe it should be "floating" instead of "fluctuating".
Corrected
- Lines 80-82: What is "expected"? This first half of the final sentence is barely understandable, please rewrite this part.
We rewrote it clearly.
Materials and Methods: Apart from a few minor issues, this section contains the necessary information, especially the description of IHC seems to be very detailed.
- Line 87: Correct to singular "Sample".
Corrected
- Line 90: Correct the unit to "g".
Corrected
- Line 96: How were the fish euthanized?
Information about the method of euthanasia was added at the beginning of the histological and histochemical paragraph.
- Line 115: Change "were" to singular "was".
Corrected
- Line 121: Change "conquer" to "quench".
Corrected
- Lines 125-131: I believe there is no need to indicate the full information about the producers repeatedly in each parenthesis. Instead, it may be grouped in several ways.
This issue was revised and the antibodies were grouped according to their producers.
- Line 145: Change "as a trial" to "in an attempt".
Corrected
Results: The quality of the presented Figure panels is on a satisfactory level. However, I spotted a few ambiguities within the written part of this section, so please pay attention to these details during the review.
- Line 156: What is "that open in the blood vessels" supposed to mean? There must be a mistake. Please correct.
We deleted it.
- Line 168: Again, the ellipsoids "open to blood vessels"?
Corrected
- Line 184: Delete "(Fig. 2A, B)".
We deleted it.
- Line 185: Delete "(Fig. 2C)" and correct to "that typically surrounded ellipsoids".
Deleted and corrected.
- Line 193: "Capsules" of rodlet cells? What is that?
Each rodlet cell is surrounded by a capsule and characterized by the presence of cytoplasmic inclusion. Please, see these references about this topic:
Mokhtar, D.M.; Abdelhafez, E.A. An overview of the structural and functional aspects of immune cells in teleosts. Histol. Histopathol.2021, 36, 399–414, doi:10.14670/HH-18-302.
Dezfuli, B.S.; Capuano, S.; Manera, M. A description of rodlet cells from the alimentary canal of Anguilla anguilla and their relationship with parasitic helminths. J. Fish Biol.1998, 53, 1084–1095, doi:10.1111/j.1095-8649.1998.tb00465.x.
- Line 206: Correct to "singularly".
Corrected.
- Line 246: Correct to "pulp".
Corrected
- Line 254: A pity that the Authors did not study a proliferation marker using IHC, such as PCNA. This would allow for a more firm identification of cellular proliferation.
As correctly highlighted by the Reviewer, we know that PCNA is a specific and useful marker of cell proliferation. Unfortunately, this marker is unavailable in our lab. So, we used Sox-9 (a specific marker of stem cells) instead of it. Thank you for this comment, we would like to add this marker to our lab in the future.
- Line 260: I believe "cells" is missing in front of "lining".
We added it.
- Line 291: Correct to "Fig. A".
Corrected.
- Line 308: Correct to plural "nuclei".
Corrected
- Lines 322-323: Correct word order to "vesicles containing pigments".
Corrected
Discussion: Overall, all findings described in the "Results" section were commented upon, what is definitely a good information. Furthermore, the English language in this section seems to be on a higher level than in other parts of the manuscript, but I still advise a full linguistic review of the entirety of the text.
- Line 325: Delete the starting sentence.
We deleted it.
- Line 329: Change "this category" to "these animals".
We changed it.
- Line 348: "Increased trend"? What trend? Please rephrase.
Corrected.
- Line 362: Again, what does "open in the blood vessels" mean in the context of ellipsoids?
We deleted it.
- Line 363: "A few species do not have ellipsoids" - what species? Citation?
Sorry for the omission, we provided to add the related species, with the citation to support this sentence. You can find it at line 400 in the new version of the manuscript.
Conclusions: Well written, comprehensive, no objections here. Well done.
Reviewer 2 Report
Comments:
The manuscript biology-1722065 provides information about structural and functional aspects of the spleen in molly fish Poecilia sphenops (Valenciennes, 1846): Synergistic interactions 3 of stem cells, neurons, and immune cells. It is an exciting work, and the ms is well written. However, the current form of ms is not suitable for publication yet as some minor issues are found in the text. The ms should be improved by addressing those issues before being accepted for publication.
Introduction
Ln 82- should be “aimed”
Materials and Methods
Ln 90- a) Did the authors evaluate the health status of fish before conducting analyses since it might affect to the results of this study? The health of fish can be determined by the assessment of stress response, highly sensitive detection of pathogens (parasite, bacteria, etc.), etc.? State this!
b) Did the author euthanize the fish before dissection? State it in the ms!

Author Response
Reviewer 2
The manuscript biology-1722065 provides information about structural and functional aspects of the spleen in molly fish Poecilia sphenops (Valenciennes, 1846): Synergistic interactions 3 of stem cells, neurons, and immune cells. It is an exciting work, and the ms is well written. However, the current form of ms is not suitable for publication yet as some minor issues are found in the text. The ms should be improved by addressing those issues before being accepted for publication.
We thank the Reviewer for the time spent carefully reviewing our manuscript. We have considered all your important comments and corrected them in the new version of the manuscript with all the requested changes. Thank you for your help. The Authors
Introduction
Ln 82- should be “aimed”
Corrected
Materials and Methods
Ln 90- a) Did the authors evaluate the health status of fish before conducting analyses since it might affect to the results of this study? The health of fish can be determined by the assessment of stress response, highly sensitive detection of pathogens (parasite, bacteria, etc.), etc.? State this!
As correctly suggested by the Reviewer, we added a paragraph in the sample collection section regarding the status of fish before conducting analyses. Thank you for this valuable comment.
- b) Did the author euthanize the fish before dissection? State it in the ms!
The method of euthanasia was added at the beginning of the histological and histochemical sections, sorry for the previous omission, this information was necessary as highlighted by the Reviewer.
Reviewer 3 Report
Overall, this paper thoroughly describes the cellular components of the spleen in the molly fish. They identified a variety of immune cell types which suggests their potential function in the immune system. The experiments were well designed and executed. Data nicely support the conclusions. It would be beneficial for the audience who are interested in the fish immune systems.
Author Response
Reviewer 3
Overall, this paper thoroughly describes the cellular components of the spleen in the molly fish. They identified a variety of immune cell types which suggests their potential function in the immune system. The experiments were well designed and executed. Data nicely support the conclusions. It would be beneficial for the audience who are interested in the fish immune systems.
Thank you very much for your comment, we gladly appreciate your opinion and support. The Authors